# Research on High Sensitivity Oil Debris Detection Sensor Using High Magnetic Permeability Material and Coil Mutual Inductance

**DOI:** 10.3390/s22051833

**Published:** 2022-02-25

**Authors:** Chengjie Wang, Chenzhao Bai, Zhaoxu Yang, Hongpeng Zhang, Wei Li, Xiaotian Wang, Yiwen Zheng, Lebile Ilerioluwa, Yuqing Sun

**Affiliations:** School of Marine Engineering, Daliann Maritime University, Dalian 116026, China; wangcj@dlmu.edu.cn (C.W.); baichenz@163.com (C.B.); 17530888625@163.com (Z.Y.); dmuliwei@dlmu.edu.cn (W.L.); wxtloveslife@dlmu.edu.cn (X.W.); evezyw@163.com (Y.Z.); lebilei7@gmail.com (L.I.); sunyq@dlmu.edu.cn (Y.S.)

**Keywords:** dual planar coils, coil mutual inductance, debris detection, microfluidic chip

## Abstract

Metallic contaminants (solid) are generated by friction pair, causing wear of equipment by enters the lubricating system. This poses a great potential threat to the normal operation of such machines. The timely analysis and detection of debris can lead to the avoidance of mechanical failures. Abnormal wear in machinery may produce debris exceeding 10 μm. The traditional inductance detection method has low sensitivity and cannot meet the actual detection requirements. To boost the sensitivity of the inductance sensor, the mutual inductance of coils and the strong magnetic conductivity of permalloy was utilized to design a high sensitivity inductance sensor for the detection of debris in lubricating oil. This design was able to detect 10–15 μm iron particles and 65–70 μm copper particles in the oil. The experimental results illustrate that low-frequency excitation is the best for detecting ferromagnetic particles, while high-frequency excitation has the best effect for detecting non-ferromagnetic particles. This paper demonstrates the significant advantages of coil mutual inductance, and strong magnetic conductivity of permalloy in improving the detection sensitivity of oil debris sensors. This will provide technical support for wear detection in mechanical equipment and fault diagnosis.

## 1. Introduction

Debris are inevitably generated during the operation of mechanical equipment, which enters the oil circulation system, affecting the normal operation of the equipment. Relevant studies have shown that the properties of debris in the oil are closely related to the friction conditions of equipment [1]. The materials used in different parts of machinery and equipment are different. Thus, by identifying the material properties of the debris, the location where the debris is generated can be roughly deduced. Information on the shape, size, and concentration of the debris reflects the attrition of mechanical equipment [2,3]. This information can be obtained to monitor the operating conditions of the equipment in real-time and to predict the occurrence of failures [4,5,6]. The basic methods currently used in oil detection include optical method [7,8], ultrasonic method [9,10,11], capacitance method [12,13], and inductance method [14,15,16]. The optical method is susceptible to light transmission and cannot distinguish between the properties of metal particles [17,18]. The ultrasonic method is easily affected by background noise and oil temperature fluctuations [19]. The capacitive method is unable to distinguish the properties of debris [20], while the inductive method can distinguish between ferromagnetic and non-ferromagnetic metal particles. The inductive method is cost-effective, simple, and easy to install. Therefore, the inductive method is often used to detect metal debris in oil [21,22].

Methods commonly used to improve the detection sensitivity of sensors include adopting a structure that can form a high gradient magnetic field inside the coil [23,24], increasing the number of excitation coils or sensing coils [25,26,27], and adding magnetic powder inside the sensing coil as a magnetic core [28,29]. Qian et al. designed a two-stage auto-asymmetrical compensation circuit based on a triple-coil debris sensor to achieve ultra-sensitive detection [30]. Currently, the 3D solenoid coil is often used as an inductive sensor due to its large sensing area, which facilitates the development of high magnetic permeability structures [23,24,29]. However, the detection accuracy of the solenoid coil decreases significantly as the diameter of the detection channel increases, and the changes in inductance signal caused by the metal particles passing through the 3D solenoid coils are significantly smaller than that of the planar coils [31,32]. Du et al. designed a double-layer planar coil as the detection element that can detect up to 50 μm iron particles [15]. Guo et al. designed three planar coils with an ampere-turn ratio of 15/7/15 in series-wound at equal intervals on the tubing. This design can detect 100 μm iron particles and 100 μm copper particles [33]. To further improve the detection accuracy of the planar coil sensor, Ma et al. added an iron core to the inner hole of the single-layer planar coil and utilized the high magnetic permeability of the iron core to increase the magnetic field strength at the sensing region, which can detect 40 μm iron particles and 90 μm copper particles [22]. Shi et al. recently designed a sensor that integrates resistance and capacitance sensing units, enhancing the sensor’s sensitivity. The sensor can detect and distinguish 16–500 μm iron particles and 60–500 μm copper particles in the oil [34].

The generation of 50–100 μm debris is a sign of failure for automobiles, airplanes, engines, and hydraulic systems. However, for some critical systems, the generation of debris exceeding 10 μm may cause system failure as it is abnormal wear for the system [35,36]. The above methods have improved the detection accuracy to a certain extent, but there is still potential for the improvement of the lower detection limit. At present, the sensor designed by Shi et al. has raised the detection limit of iron particles and copper particles to 16 μm and 60 μm, respectively. Nevertheless, this does not meet the detection requirements for some critical systems. Therefore, it is necessary to improve further the detection limit of metal particles based on the previous research. In this paper, a sensor, composed of two planar coils connected in series completely wrapped by permalloy, is designed. This structure takes advantage of the mutual inductance between the coils and the strong permeability of permalloy, to improve the sensitivity of the sensor for the detection of debris.

## 2. Sensor Design and Manufacturing Process

The 3D image of the sensor is shown in Figure 1a, which is composed of a glass slide, matrix, sensing unit, detection microchannel, inlet, and outlet. Figure 1b,c are the expanded view of the sensing unit and the schematic diagram showing the size of the permalloy, respectively. The sensing unit consists of two single-layer planar coils and three permalloy sheets. Permalloy has high permeability, low coercivity, high saturation magnetization, sensitive response to the weak magnetic signal, and good magnetic shielding function. Firstly, the sensing coil was wrapped by permalloy, which acts as a magnetic shield; this process significantly reduces any external electromagnetic interference. Then, the high magnetic permeability of permalloy converges the coil’s magnetic field towards the center of the coil, thereby enhancing the magnetic field in the detection region of the coil [37,38].

When fabricating the sensor, a circular hole of 2.2 mm in diameter bore on one of the permalloys, and a circular hole of 0.8 mm in diameter was drilled on the other two permalloy sheets. The two planar coils are glued coaxially and embedded on the permalloy sheet, which bore a 2.2 circular hole. Then, the other two permalloy sheets were attached to both sides of the permalloy sheet. This arrangement ensures that the round hole of permalloy and the inner hole of the planar coil is coaxial.

Previous studies revealed that the more the turns coil, the greater the detection signal and noise value. Furthermore, the optimum detection can be obtained when a coil of 20 turns is utilized [39]. Therefore, to ensure that the total number of turns used is 20 turns, two 10-turn single-layer planar coils were used in series. The inner and the outer diameter of each planar coil is 0.4 mm and 2.0 mm, respectively. Copper wire with a diameter of 0.07 mm and covered with a 5 μm thick insulating varnish was used to wind the coil. Figure 1d shows the physical picture of the sensor’s induction unit. The planar coil terminals are welded on four copper sheets, and each sheet is connected to a wire.

## 3. Detection Principle and Simulation Analysis

As shown in Figure 2, an orthogonal coordinate system $\;\left(X,Y,Z\right)$ was established. The axis of the coil coincides with the *X*-axis, and the center of the axis is at the point O. To simplify the calculation, a cylindrical coordinate system $\;\left(\rho ,\;\varphi ,\;Z\right)$ is established based on this orthogonal coordinate system, where the point O is the origin of the polar axis, and the *Z*-axis is the polar axis. The metal particles get magnetized by the magnetic field $\left(B_{P}\right)$ generated by the coil at the point P. The orthogonal coordinate system $\left(X_{P},Y_{P},Z_{P}\right)$ takes the center of the metal particle as the origin and $Z_{P}$ is parallel to $B_{P}$.

According to previous research [40], the change in the magnetic vector potential at the point O caused by the metal particles at the point Q is:(1)$$\Delta A\left(Q\right)=\frac{1}{4\pi }v\chi_{a}B\left(P\right)\frac{r_{Q}-r_{P}}{{\left|r_{Q}-r_{P}\right|}^{3}}$$
where v is the volume of the metal particle, $\chi_{a}$ is the magnetic susceptibility of the metal particle, $B\left(P\right)$ is the magnetic field generated by the two planar coils in series at the point P, $r_{Q}$ is the distance of the point Q from the coordinate origin O, $r_{P}$ is the distance of point P from the coordinate origin O.

The magnetic susceptibility $\chi_{a}$ of metal particles in a time-harmonic magnetic field is given:(2)$$\chi_{a}=\frac{3}{2}\frac{\left(a^{2}\lambda^{2}-2\mu_{r}-1\right)\sin\left(a\lambda \right)+a\lambda \left(2\mu_{r}+1\right)\cos\left(a\lambda \right)}{\left(a^{2}\lambda^{2}+\mu_{r}-1\right)\sin\left(a\lambda \right)-a\lambda \left(\mu_{r}-1\right)\cos\left(a\lambda \right)}$$
where a is the radius of the metal particles, $\mu_{r}$ is the relative permeability and the value of λ can be obtained:(3)$$\lambda =\sqrt{-j\omega \mu_{r}\mu_{0}\sigma }$$
where j is an imaginary unit, ω is the angular frequency of the excitation current in the coil, $\mu_{0}$ is the vacuum permeability, and σ is the conductivity.

The changes in coil impedance caused by metal particles are:
(4)$$\Delta {\tilde{Z}}_{m}=j\omega n_{c}\oint_{V_{c}}\left[\frac{\Delta A(Q)}{I}\right]dV\left(Q\right)=j\omega n_{c}\int_{0}^{2\pi }d\varphi_{Q}\int_{R_{1}}^{R_{2}}\rho_{Q}d\rho_{Q}\int_{-D}^{D}\left[\frac{\Delta A(Q)}{I}\right]dz_{Q}$$

Metal particles causes eddy current when passing through the induction area. The eddy current will generate a magnetic field opposite to the coil’s magnetic field, as shown in Figure 3a. The changes of this magnetic field will be measured by the sensing coil. Figure 3b is the equivalent circuit diagram of the eddy current detection. At this time, the detected object is regarded as a short-circuited coil.

According to Kirchhoff’s law:(5)$$\left\{\begin{array}{c} \dot{U}=\left(R_{1}+j\omega L_{1}\right){\dot{I}}_{1}-j\omega M{\dot{I}}_{2}\\ 0=\left(R_{2}+j\omega L_{2}\right){\dot{I}}_{2}-j\omega M{\dot{I}}_{1} \end{array}\right.$$

Solving the above equations:(6)$${\dot{I}}_{1}=\frac{{\dot{U}}_{1}}{\left[R_{1}+\frac{{\left(2\pi f\right)}^{2}M^{2}}{R_{2}^{2}+{\left(2\pi fL_{2}\right)}^{2}}R_{2}\right]+j\left[2\pi fL_{1}-\frac{{\left(2\pi f\right)}^{2}M^{2}}{R_{2}^{2}+{\left(2\pi fL_{2}\right)}^{2}}2\pi fL_{2}\right]}$$
(7)$${\dot{I}}_{2}=\frac{M{\left(2\pi f\right)}^{2}L_{2}L_{1}+j2\pi fMR_{2}I_{1}}{R_{2}^{2}+{\left(2\pi fL_{2}\right)}^{2}}$$

Therefore, the equivalent impedance of the eddy current effect sensing coil is:(8)$$\Delta {\tilde{Z}}_{c}=\frac{{\dot{U}}_{1}}{{\dot{I}}_{1}}=R_{1}+\frac{{\left(2\pi f\right)}^{2}M^{2}}{R_{2}^{2}+{\left(2\pi f\right)}^{2}L_{2}^{2}}R_{2}+j\left[2\pi fL_{1}-\frac{{\left(2\pi f\right)}^{3}M^{2}L_{2}}{R_{2}^{2}+{\left(2\pi f\right)}^{2}L_{2}^{2}}\right]$$

From the above formula, the expression of the inductance change is:(9)ΔL=ImΔZ˜ω=ImΔZ˜m+ΔZ˜cω

To investigate the effectiveness of the sensor design, COMSOL Multiphysics 5.4 software was used to simulate and analyze the magnetic field in the sensing area. The model structure and material of the COMSOL simulation analysis were consistent with the designed sensor. The microchannel was filled with hydraulic oil, and the excitation frequency and voltage were set as 2.0 MHz and 2.0 V.

Figure 4 shows the simulation results of the magnetic field in the sensing area. Figure 4a,b are schematic diagrams of the simulation area. The radial direction of the coil is defined as the *X*-axis, and the axial direction of the coil is the *Y*-axis. The range of the simulation area is X (1.75–2.25 mm), Y (0–0.66 mm), and the direction of the flow channel is parallel to the direction of the coil axis.

Figure 4c is a 3D-color mapping surface diagram of the magnetic field intensity distribution in the sensing area under different structures. The three curved surfaces from top to bottom correspond to the three situations of Figure 4e–g, respectively. Comparing the upper and lower surfaces in Figure 4c with comparing Figure 4e,g, it can be seen that after the detection coil is completely wrapped by permalloy, the magnetic field intensity in the sensing area is significantly enhanced. Likewise, comparing the upper and middle surfaces in Figure 4c,e,g, it can be seen that mutual inductance will be generated when two planar coils are connected in series, which can also significantly increase the magnetic field strength of the sensing area.

Figure 4d illustrates the magnetic field distribution in the radial and axial directions of the coil. It can be seen that the magnetic field strength in the axial middle position (y = 0.33 mm) is the highest, and the magnetic field strength in the radial middle position (x = 2 mm) is the lowest. The rate of change of the axial magnetic field strength is greater than that of the radial direction. The rate of change of the magnetic field strength in both directions is at the highest when the two planar coils in series are wrapped by permalloy, while the rate of change of magnetic field strength is at the lowest when the two planar coils are connected in series without permalloy.

## 4. Experiment and Data Analysis

The schematic diagram of the experimental system is presented in Figure 5, in which the designed microfluidic detection sensor is the most important part. The coil of the sensor was connected to the LCR meter (Agilent E4980 A, Agilent Technologies Inc., Bayan Lepas, Malaysia), which was used to excite the detection coil. The LCR meter was also connected to a computer with LabVIEW software to collect the detection data; when metal debris passes through the sensing area, the inductance value of the planar coil will change, and this change value will be collected by the LCR meter. The collected data was saved to the computer through LabVIEW.

Before the experiment, the iron particles and copper particles of different sizes were mixed with tested hydraulic oil (The Great Wall L-HM 46, Sinopec Lubricant Co., Ltd., Beijing, China). A syringe installed on the micro-syringe pump (Harvard Apparatus B-85259, Harvard Apparatus, Holliston, MA, USA) was filled with an appropriate amount of oil sample to be tested, then connected to inlet of the detection channel of the sensor. The speed of the micro-injection pump was set as 40 μL/min. A computer was connected to a microscope (Nikon AZ100, Nikon, Tokyo, Japan) to observe the particles passing through the sensing area. During the experiment, the excitation voltage and frequency of the LCR meter were set as 2 V and 2 MHz, respectively. Then, the detection mode was set to inductance mode.

To observe the effect of the sensor, two sensors were used to experiment. Both sensors have two 10-turns positive series planar coils, but one was wrapped with permalloy and the other without permalloy. The two sensors were used to test the oil samples containing different abrasive particles.

Figure 6a shows the waveform plots for detecting 80–90 μm iron particles, while Figure 6b shows the waveform plots for detecting 140–150 μm copper particles. The results illustrate that the detection effect is significantly improved after the coil has been completely wrapped by permalloy, with the signal amplitude increasing by around 50% for iron particles and 156% for copper particles, while the noise value remains largely unchanged as shown in Figure 6a,b, respectively. This means that using a permalloy-wrapped coil structure can increase the sensing area’s magnetic field intensity by utilizing the permalloy’s magnetic permeability. This result is consistent with the simulation results.

In order to study the influence of different winding methods of induction coils, a comparison experiment was carried out in this study between two 10-turn single-layer planar coils connected in series and 20-turn double-layer planar coils. Figure 7a shows the structure diagram of two single-layer planar coils. During the experiment, the points B and C are connected in a manner that the two coils are in series, and A–B–C–D represents the current flow direction. Figure 7b shows the structure diagram of the double-layer planar coil, and a–b is the current flow direction.

The simulation results show that after two single-layer planar coils are connected in series, the magnetic field intensity in the induction zone is significantly enhanced.

Figure 8a shows the waveform plots for detecting 80–90 μm iron particles, while Figure 8b shows the waveform plots for detecting 140–150 μm copper particles. As shown in Figure 8, the experimental results indicate that the mutual inductance generated by the coils in series increases the detecting the signal amplitude by around two times, while the noise value is unchanged. Thereby indicating that the mutual inductance between the coils can effectively improve the detection accuracy of the sensor, which is consistent with the simulation results.

To study the influence of the excitation frequency on the detection, the excitation applied on the coil by the impedance analyzer was adjusted, and the oil metal particle detection experiment was carried out.

Figure 9a,b show the detected signal value, noise value, and SNR of 50–60 μm iron particles and 100–110 μm copper particles at different frequencies. As shown in Figure 9a, the signal-to-noise ratio for ferromagnetic metal particles attains the highest value when the excitation frequency is 0.4 MHz. For non-ferromagnetic metal particles, the detection signal amplitude and SNR increase with the increase in the excitation frequency, while the noise value decreases with the increase in the excitation frequency, as indicated in Figure 9b.

When the excitation frequency was increased from 1.0 MHz to 2.0 MHz, the signal amplitude of detecting 100–110 μm copper particles continues to increase from 4.36 × 10^−11^ H to 8.33 × 10^−11^ H, and the noise value continues to decrease from 1.48 × 10^−11^ H to 1.00 × 10^−11^ H, while SNR continues to increase. It can be seen from the experimental results that the optimal frequency of the designed chip for the detection of ferromagnetic metal particles is 0.4 MHz. In contrast, the detection of non-ferromagnetic metal particles is suitable for higher-frequency excitation.

Through the above analysis, it can be observed that the detection signal of the sensor with having two 10-turn single-layer planar coils in series, wrapped by permalloy is effective, and the optimal excitation frequency of different metal particles is different. Therefore, 0.4 MHz excitation is used for the lower detection limit experiment of iron particles, and the 2 MHz excitation is chosen for the lower detection limit experiment of copper particles. The research results show that the designed chip can detect 10–15 μm iron particles and 65–70 μm copper particles as shown in Figure 10a,b, respectively. Additionally, Figure 10a,b are the detection signal diagrams of 10–15 μm iron particles and the detection signal diagrams of 65–70 μm copper particles.

## 5. Conclusions

In this paper, a high-sensitivity oil debris detection sensor is proposed. The sensing unit of the sensor consists of two series planar coils wrapped by permalloy. The results show that the detection sensitivity can be improved by connecting two planar coils in series and wrapping them with permalloy. Better results can be obtained by using different excitation frequencies for different metal debris detection: low-frequency excitation is suitable for detecting ferromagnetic debris, while high-frequency excitation is suitable for detecting non-ferromagnetic debris. The debris detection sensor designed in this research can detect 10–15 μm iron particles and 65–70 μm copper particles under the optimal excitation frequency. The limit of detection using an inductive sensor is further improved in this paper. Later, the structure of the sensing unit will be optimized to improve the detection throughput and also ensure high detection sensitivity. Some noise reduction filtering techniques will be used to improve the detection sensitivity and robustness of the sensor. This work provides a technical scheme for oil condition monitoring and fault diagnosis of equipment, which can prevent the occurrence of equipment failure and prolong its service life.

## Figures and Tables

**Figure 1 sensors-22-01833-f001:**
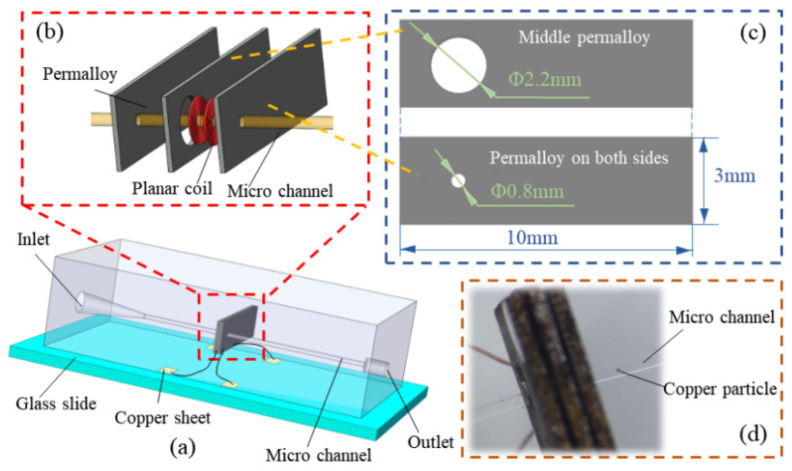
Schematic of the oil debris sensor: (**a**) three-dimensional structure of the sensors, (**b**) exploded view of induction unit, (**c**) size of the permalloy, (**d**) physical map of sensing unit.

**Figure 2 sensors-22-01833-f002:**
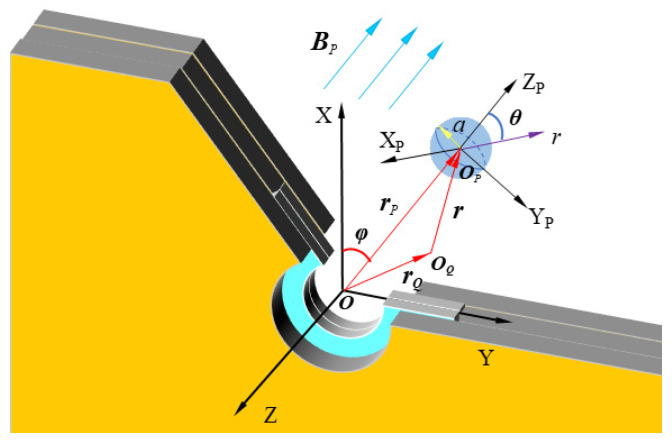
Magnetic field on the circular current-carrying conductor.

**Figure 3 sensors-22-01833-f003:**
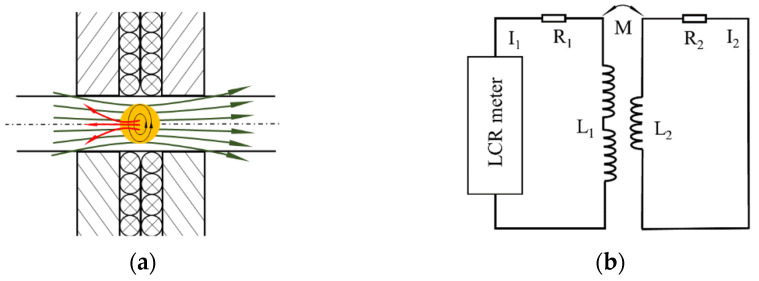
Eddy current effect of (**a**) eddy current coupling model of metal particles and coils, (**b**) eddy current detection equivalent circuit.

**Figure 4 sensors-22-01833-f004:**
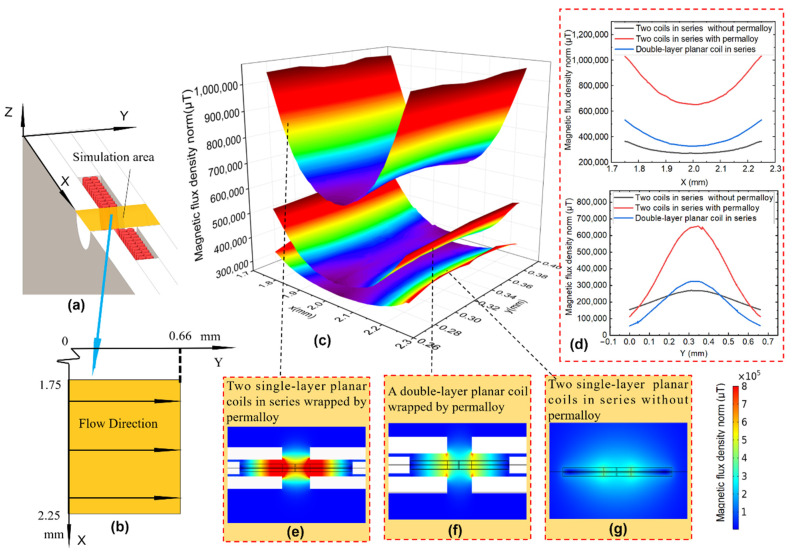
Magnetic field simulation results: (**a**) simulation area, (**b**) simulation area boundary, (**c**) magnetic flux density norm of different structure, (**d**) radial and axial magnetic field distribution of the coil, (**e**) magnetic field distribution diagram of two planar coils in series wrapped by permalloy, (**f**) magnetic field distribution diagram of a double-layer planar coil wrapped in permalloy, (**g**) magnetic field distribution diagram of two planar coils in series that are not wrapped by permalloy.

**Figure 5 sensors-22-01833-f005:**
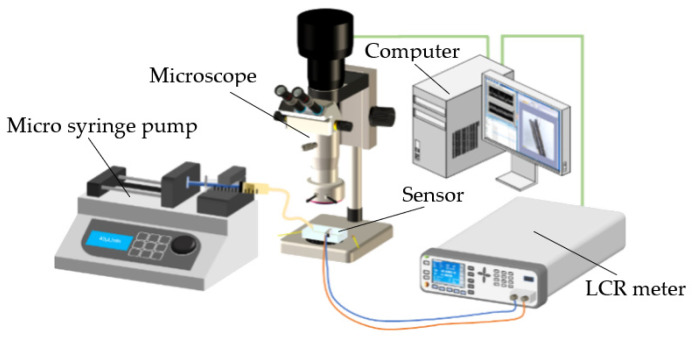
Experimental Systems.

**Figure 6 sensors-22-01833-f006:**
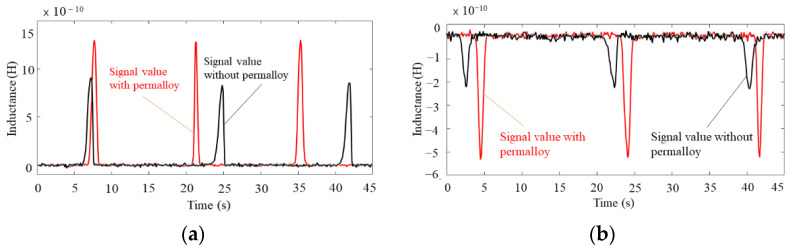
Waveform diagram of metal particle detection with or without permalloy package: (**a**) waveform plots for detecting 80–90 μm iron particles, (**b**) waveform plots for detecting 140–150 μm copper particles.

**Figure 7 sensors-22-01833-f007:**
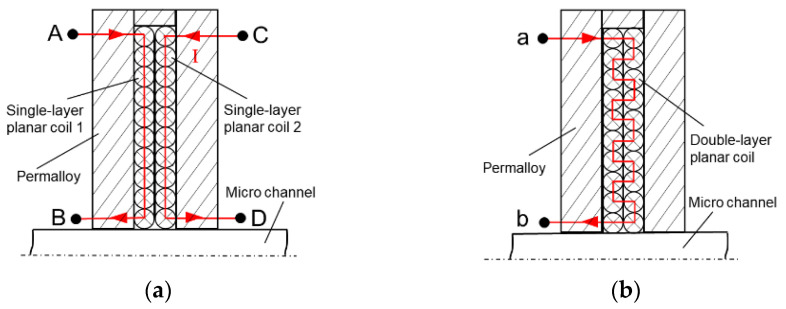
The difference between the two winding methods: (**a**) two planar coils in series, (**b**) double-layer planar coil.

**Figure 8 sensors-22-01833-f008:**
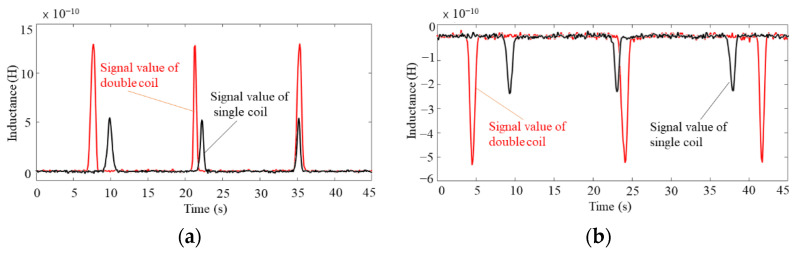
Waveform diagram of metal particle detection with or without permalloy package: (**a**) waveform plots for detecting 80–90 μm iron particles, (**b**) waveform plots for detecting 140–150 μm copper particles.

**Figure 9 sensors-22-01833-f009:**
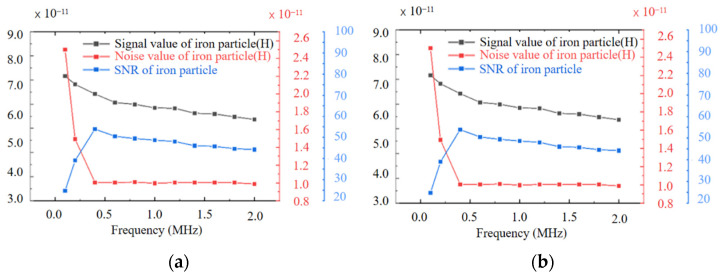
Detection signal value, noise value, and SNR of metal particles under different excitation frequencies: (**a**) waveform plots for detecting 50–60 μm iron particles, (**b**) waveform plots for detecting 100–110 μm copper particles.

**Figure 10 sensors-22-01833-f010:**
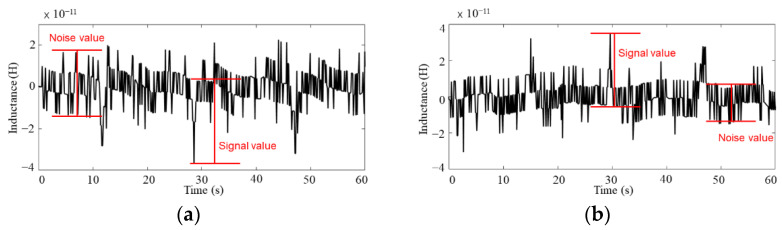
Waveform diagram of metal particle detection with or without permalloy package: (**a**) waveform plots for detecting 10–15 μm iron particles, (**b**) waveform plots for detecting 65–70 μm copper particles.

## Data Availability

Not applicable.

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
