# Peer review of "Research on High Sensitivity Oil Debris Detection Sensor Using High Magnetic Permeability Material and Coil Mutual Inductance"

_sensors, 2022, doi:10.3390/s22051833_

Round 1

Reviewer 1 Report

The manuscript is a good documentation of the sensor design and its simulation and testing results. The missed pieces as a journal article are rationales behind choices made in the sensor design and simulation (please check the detailed comments below on the specifics). For literature review in the introduction section, please consider to add some work from America or Europe or other parts of the world. Some of the work cited in the literature review section appears to be previous work of one or a few authors for the current manuscript. Some detailed comments are provided below. Please revise and resubmit.

A few detailed comments:

  1. page 2, lines 88-89, how is the architecture determined? what is the rationale?
  2. page 3, figure 1, please check the caption.
  3. page 3, line 103, please make sure the citation formatting is correct. Currently, [15]-[41] implies both and all in between.
  4. page 3, lines 105 to 107, how are these parameters chosen? why?
  5. page 4, line 115, particles will be generated does not read right.
  6. page 4, line 128, please check the definitions. It appears j is defined twice.
  7. page 4, line 137, please correct the phrase.
  8. page 5, line 149, the unit of mm could be added to an appropriate spot.
  9. page 5, line 154, figure 5 might be figure 4.
  10. page 5, figure 4, do the results shown in (d) correspond to (e), (f), (g)? If not, please consider to harmonize.
  11. page 9, figure 10, caption should be on 10-15 µm and 65-75 µm.

Author Response

Dear reviewer,

On behalf of my co-authors, we thank you very much for giving us an opportunity to revise our manuscript. We have studied your valuable comments carefully and tried our best to revise our manuscript according to the comments. The references section has been revised accordingly. The modified content has been marked with yellow in the manuscript and the responses to your comments is as follows:

  1. page 2, lines 88-89, how is the architecture determined? what is the rationale?

Reply: Thank you very much for your valuable comments. This structure is adopted because: firstly, permalloy has strong magnetic permeability, and the use of permalloy to completely wrap the detection coil can make the magnetic field of the planar coil gather to the induction area; secondly, after the two planar coils are connected in series, there will be mutual inductance. The existence of mutual inductance can significantly enhance the magnetic field in the sensing area and improve the detection performance of the sensor. The above conclusions can be verified by finite element analysis and experiments.

  1. page 3, figure 1, please check the caption.

Reply: Thank you very much for your valuable comments. I have carefully checked the caption of Figure 1, because of my negligence there are indeed problems, which has been modified in the manuscript.

  1. page 3, line 103, please make sure the citation formatting is correct. Currently, [15]-[41] implies both and all in between.

Reply: Thank you very much for your valuable comments. I have carefully checked the manuscript, and there are some mistakes with the citation formatting, which has been modified in the manuscript.

  1. page 3, lines 105 to 107, how are these parameters chosen? why?

Reply: Thank you very much for your valuable comments. In our previous research, we found that the detection effect is the best when the number of turns of the planar coil is 20 turns [39]. So the number of turns of the coil after the series connection is set to 20 turns, and each planar coil is 10 turns; 0.07mm copper wire is often used for winding the coil, and the copper wire is wrapped with 0.005mm insulating paint. This type of enameled wire is also used in this study. The microchannel of the sensor is made of 0.3mm copper wire as the core. In order to ensure that the PDMS completely enters the inside of the coil, the inner diameter of the planar coil is selected to be 0.4mm. The above contents have been added in the manuscript.

Reference

[39] Bai, C. Zhang, H. Zeng, L. Zhao, X. and Yu, Z. High-throughput sensor to detect hydraulic oil contamination based on microfluidics. IEEE Sensors Journal. 2019, 19, 8590-8596.

  1. page 4, line 115, particles will be generated does not read right.

Reply: Thank you very much for your valuable comments. The description here has some problems. In the manuscript, it has been changed to ‘metal particles will be magnetized by the magnetic field  generated by the coil at  point’.

  1. page 4, line 128, please check the definitions. It appears j is defined twice.

Reply: Thank you very much for your valuable comments. Here is my negligence, the latter ’j’ should be ’ω’, which has been revised in the manuscript.

  1. page 4, line 137, please correct the phrase.

Reply: Thank you very much for your valuable comments. The phrase has been revised in the manuscript.

  1. page 5, line 149, the unit of mm could be added to an appropriate spot.

Reply: Thank you very much for your valuable comments. The unit of mm has been added to an appropriate spot in the manuscript.

  1. page 5, line 154, figure 5 might be figure 4.

Reply: Thank you very much for your reminder. This error has been modified in the manuscript.

  1. page 5, figure 4, do the results shown in (d) correspond to (e), (f), (g)? If not, please consider to harmonize.

Reply: Thank you very much for your valuable comments. In Fig. 4, the red line, blue line and black line in (d) correspond to the three structures in Fig. 4 (e), (f) and (g), respectively.

  1. page 9, figure 10, caption should be on 10-15 µm and 65-75 µm.

Reply: Thank you very much for your reminder. This error has been modified in the manuscript.

Thank you again for your valuable comments on the improvement of our manuscript! Best wishes to you!

Reviewer 2 Report

In my point of ,the proposed approach is a good solution for high sensitivity oil debris detection.

The only thing I concern is the novelty of this manuscript, the similar results has been report in the follwing paper.

 An impedance debris sensor based on a high-gradient magnetic field 
for high sensitivity and high throughput. IEEE Transactions on Industrial Electronics. 2021, 68, 5376-5384

Author Response

Dear reviewer,

On behalf of my co-authors, we thank you very much for giving us an opportunity to revise our manuscript. Below is a response to your comments:

  1. The only thing I concern is the novelty of this manuscript, the similar results has been report in the follwing paper.

 An impedance debris sensor based on a high-gradient magnetic field for high sensitivity and high throughput. IEEE Transactions on Industrial Electronics. 2021, 68, 5376-5384

Reply: Thank you very much for your valuable comments. the novelty of this manuscript is:

(1) This paper studies the influence of different winding methods of the planar coil on the detection performance of the sensor;

(2) This paper innovatively uses a structure that uses permalloy to completely wrap the plane coil, which can more effectively improve the detection accuracy of the sensor;

(3) The detection limit of this paper is further improved. The paper you mentioned can detect iron particles above 25µm and copper particles above 85µm, while the sensor designed in this study can detect iron particles of 10-15µm and copper particles of 65-70µm. The sensitivity of the sensor is significantly improved.

Thank you again for your valuable comments on  our manuscript! Best wishes to you!

Reviewer 3 Report

The authors proposed a sensor of high magnetic permeability material to detect oil debris from coil mutual inductance. It's quite an interesting topic and the method is presented very well in this paper with solid experimental validation. There are some aspects that may help to improve the paper further.

  1. Please consider how to implement this method on-line, as the inductance is measured using a LCR meter which is not a cost-effective approach for field-deployment.
  2. Could you please analyse how debris would affect the magnetic field in COMSOL simulation, and will it give you the same results as the analytical analysis and experiments.

Author Response

Dear reviewer,

On behalf of my co-authors, we thank you very much for giving us an opportunity to revise our manuscript. Your comments is very important to us, and the responses to your comments please see the attachment.

Thank you again for your valuable comments on our manuscript! Best wishes to you!
